# *Trichoderma*: Dual Roles in Biocontrol and Plant Growth Promotion

**DOI:** 10.3390/microorganisms13081840

**Published:** 2025-08-07

**Authors:** Xiaoyan Chen, Yuntong Lu, Xing Liu, Yunying Gu, Fei Li

**Affiliations:** 1School of Life Sciences, Guizhou Normal University, Guiyang 550025, China; chenxy8387@126.com (X.C.); 18984028920@163.com (Y.L.); 222100100431@gznu.edu.cn (X.L.); 15718669671@163.com (Y.G.); 2Key Laboratory for Information System of Mountainous Area and Protection of Ecological Environment of Guizhou Province, Guizhou Normal University, Guiyang 550025, China; 3State Key Laboratory of Microbial Technology, Shandong University, Qingdao 266237, China

**Keywords:** *Trichoderma*, sustainable agriculture, biocontrol agent, plant protection, evolution and adaptation

## Abstract

The genus *Trichoderma* plays a pivotal role in sustainable agriculture through its multifaceted contributions to plant health and productivity. This review explores *Trichoderma*’s biological functions, including its roles as a biocontrol agent, plant growth promoter, and stress resilience enhancer. By producing various enzymes, secondary metabolites, and volatile organic compounds, *Trichoderma* effectively suppresses plant pathogens, promotes root development, and primes plant immune responses. This review details the evolutionary adaptations of *Trichoderma*, which has transitioned from saprotrophism to mycoparasitism and established beneficial symbiotic relationships with plants. It also highlights the ecological versatility of *Trichoderma* in colonizing plant roots and improving soil health, while emphasizing its role in mitigating both biotic and abiotic stressors. With increasing recognition as a biostimulant and biocontrol agent, *Trichoderma* has become a key player in reducing chemical inputs and advancing eco-friendly farming practices. This review addresses challenges such as strain selection, formulation stability, and regulatory hurdles and concludes by advocating for continued research to optimize *Trichoderma*’s applications in addressing climate change, enhancing food security, and promoting a sustainable agricultural future.

## 1. Introduction

*Trichoderma*, also known in its teleomorphic form as Hypocrea, is a genus of filamentous fungi renowned for its mycotrophic abilities, meaning it can parasitize other fungi [1]. This genus is globally distributed and thrives in a wide range of environments, including agricultural lands, forests, grasslands, deserts, and both freshwater and marine ecosystems. *Trichoderma* species are characterized by their rapid growth on various substrates and their prolific production of green conidia, which makes them easily identifiable [2].

*Trichoderma* species are considered opportunistic symbionts that are highly beneficial to plants. While the majority of *Trichoderma* species are non-pathogenic saprophytes or beneficial symbionts in soil and plant ecosystems, a few can act as opportunistic pathogens in specific contexts, such as in immunocompromised hosts or under rare environmental conditions [3,4]. They enhance plant health through multiple mechanisms. In terms of biological control of plant diseases, *Trichoderma* has direct antagonistic actions against pathogens. It can also indirectly strengthen plant defense systems via local and systemic responses. Moreover, *Trichoderma* is capable of stimulating root development and promoting plant growth. This dual-benefit situation is advantageous for both the host plant and the fungus itself, showing great potential in agriculture. The interactions between plants and *Trichoderma* are complex. Root colonization by *Trichoderma* and the bioactive compounds it produces activate biochemical and genetic pathways in plants [5]. These pathways play a crucial role in enhancing plants’ ability to defend against both biotic and abiotic stresses [6].

Recent advancements in omics technologies have further illuminated the ecological roles of *Trichoderma* species, providing insights into their evolution from common soil dwellers that decompose organic matter to mycoparasites that interact with other fungi, including those from closely related taxonomic groups [7,8]. These technological approaches have deepened our understanding of how *Trichoderma* benefits plants [9], particularly in the context of agricultural productivity. Molecular methods have significantly expanded *Trichoderma*’s taxonomy, increasing the number of recognized species from just nine fifty years ago to over four hundred today [10].

Research into *Trichoderma* has evolved to incorporate integrated, multidisciplinary approaches that explore its diverse roles as a plant-beneficial fungus [11]. As of 2025, the genus encompasses over 400 identified species, with ongoing discoveries driven by advanced genomic sequencing and phylogenetic analyses, revealing unprecedented diversity in habitats ranging from forest soils to agricultural fields (e.g., recent additions like *T. cerradensis* sp. nov. and *Trichoderma egyptiacum* sp. nov., identified through metagenomic studies in tropical ecosystems) [12,13]. This taxonomic expansion underscores *Trichoderma*’s adaptability and underscores its pivotal roles in sustainable agriculture development, including serving as a biocontrol agent to naturally suppress soilborne pathogens like *Fusarium* and *Rhizoctonia*, thereby reducing chemical pesticide use in field applications; acting as a biostimulant to enhance plant growth through improved nutrient uptake (e.g., phosphorus solubilization) and stress tolerance against drought, salinity, and heavy metals, leading to yield increases in crops such as maize and tomatoes; promoting soil health by boosting microbial diversity and organic matter decomposition; and facilitating bioremediation of contaminated soils by degrading pollutants like hydrocarbons and pesticides [3]. Consequently, *Trichoderma* is being recognized as a vital biotechnological tool in modern agricultural practices, enabling eco-friendly innovations like biofertilizers and microbiome engineering for resilient, low-input farming systems worldwide.

This review summarizes recent advancements in *Trichoderma* research, including its ecological physiology, adaptive behaviors leading to species diversification, and its dual roles in both direct biological control of pests and indirect stimulation of plant immune responses. We also explore its potential as a biostimulant for promoting plant growth and enhancing resilience to abiotic stress. Furthermore, this review discusses future strategies for improving *Trichoderma* applications, optimizing bioformulations, and considering policy implications. Ultimately, we explore how the use of *Trichoderma* could contribute to reducing chemical inputs in agriculture, fostering a more sustainable and environmentally friendly farming system.

## 2. Evolution and Ecological Adaptations of *Trichoderma*: From Saprotrophism to Plant Mutualism

The *Trichoderma* genus exhibits notable morphological consistency and nutritional diversity, with a species count surpassing other fungi in similar ecological niches [10]. This diversity stems from four major evolutionary transitions that reshaped its nutritional strategies and ecological roles, leading to increased species diversification (Figure 1). Initially, *Trichoderma* shifted from parasitizing fungi-decomposing plants to saprotrophism, feeding on decaying plant matter. It then adapted to soil environments, developing saprotrophic, mycotrophic (fungus-feeding), and phytophagic (plant-feeding) behaviors. The third transition introduced mycoparasitism, where *Trichoderma* fed on living fungi, and the fourth established symbiotic relationships with living plants.

Genomic studies reveal that *Trichoderma* species have continually adapted their genomes to colonize new environments. Mycotrophy, an ancient trait, has been crucial for its success and beneficial plant interactions. Phylogenomic analyses suggest that *Trichoderma* shares a common ancestor with entomopathogenic fungi, with its earliest species evolving around the Cretaceous–Paleogene extinction event about 66 million years ago [14]. During this time, *Trichoderma* was mycoparasitic on *Basidiomycota* hosts, acquiring genes through horizontal gene transfer, enabling growth on decaying wood [15].

As *Trichoderma* evolved, it expanded its ecological niche to include mycoparasitism of related fungi like *Ascomycota* and parasitism of soilborne pathogens such as *Phytophthora*, *Pythium*, *Rhizoctonia*, and nematodes. It also formed mutualistic relationships with insects, protecting termites from entomopathogenic fungi [16]. In soil, *Trichoderma* increased enzyme production, including exochitinases and endochitinases, enhancing mycoparasitism and its competitive advantage. Many hydrolytic enzymes originated from lateral gene transfer from plant-associated fungi.

*Trichoderma*’s adaptation to various habitats led to rhizosphere colonization, attracted by fungal prey and plant root nutrients [17]. This eventually resulted in plant tissue colonization, evolving into endophytes that live within plants [18]. This transition marked *Trichoderma*’s evolution toward intimate plant interactions, acting as a non-pathogenic mutualist that promotes plant growth and provides protection against biotic and abiotic stresses [18].

*Trichoderma* exhibits significant opportunistic traits, allowing it to thrive in a variety of ecological niches by utilizing a range of strategies to compete for space and nutrients, withstand environmental stresses, and modify its habitat to its advantage. This includes detoxifying harmful substances, altering substrate pH, and inducing changes that promote its own survival [19]. Root-derived nutrients, such as pectin, xylan, and mucigel-released compounds, serve as attractants for *Trichoderma*, driving its colonization of plant roots [20,21]. The fungus also produces reactive oxygen species (ROS), which play a role in antagonizing phytopathogens, particularly those with cellulose-rich cell walls, such as *Pythium ultimum* [22]. Additionally, ROS and oxylipins in root exudates from plants under stress, such as pathogen attack or salt exposure, act as selective chemoattractants that further enhance *Trichoderma*’s growth [23,24,25].

Beyond its role in pathogen defense, *Trichoderma* contributes to plant health by modulating the plant’s antioxidant defense systems, reducing ROS levels, and limiting tissue damage under abiotic stress [26,27]. Unlike many other fungi, *Trichoderma* possesses a robust antioxidant system that supports both its own genomic stability and the plant’s resilience to oxidative damage. Moreover, the fungus can repair DNA damage caused by UV radiation and regulate metabolic processes in response to light, highlighting its versatile ability to adapt to environmental challenges [28].

*Trichoderma* also enhances its ecological role through the secretion of various compounds, such as siderophores [29], which facilitate its competition for iron in the rhizosphere, and volatile organic compounds (VOCs), like 6-pentyl-2H-pyran-2-one, which have antibiotic properties and influence plant growth, root development, and immune responses [30]. The ability to produce phytohormones such as auxins, gibberellins, and cytokinins further links *Trichoderma* to plant growth promotion [26,31], although excessive auxin accumulation can sometimes inhibit root development by acidifying the rhizosphere [32]. Additionally, non-secreted cell wall proteins, like QID74 in *Trichoderma harzianum*, increase root hair formation, enhancing nutrient absorption and contributing to plant biomass.

The endophytic colonization of plants by *Trichoderma* plays a crucial role in both disease control and plant growth enhancement [33]. By colonizing plant tissues, *Trichoderma* provides protection against pathogens, such as *Verticillium dahliae*, and helps improve photosynthetic capacity and stress tolerance [34,35]. However, the outcomes of endophytic colonization can be inconsistent, and further research is needed to understand the conditions under which these benefits are most pronounced.

*Trichoderma* species exhibit remarkable ecological adaptability, particularly in colonizing the rhizosphere and plant root systems. Their capacity for biocontrol and antagonism highlights their competitive and aggressive nature in occupying ecological niches. Given this, it is essential to evaluate the potential impacts of *Trichoderma* on non-target organisms, including plants and soil microbial communities. Notably, *Trichoderma* is often considered an indicator of healthy soil [36]. A core *Trichoderma* population appears to exist across diverse plant species worldwide, with endemic plants generally supporting a higher abundance of antagonistic strains [14]. Agricultural practices, including crop selection and cultivation methods, significantly influence soil characteristics and fungal diversity—affecting both harmful and beneficial fungi. Typically, microbial diversity peaks in bulk soil and declines in the rhizosphere and endosphere. However, *Trichoderma* inoculation alters both bacterial and fungal populations in all these zones [37]. Its application, either alone or combined with organic compost, has been shown to support plant growth and restructure rhizosphere microbial communities, particularly by enhancing phosphorus solubilization and encouraging beneficial microbial consortia [38,39].

Furthermore, different organic soil amendments variably affect the growth and disease-suppressive abilities of microorganisms, including *Trichoderma*, and promote greater root development [40]. *Trichoderma* can also help preserve microbial diversity under adverse conditions. For instance, it has been observed to increase the richness and abundance of beneficial bacteria in wheat rhizospheres that were negatively impacted by excessive nitrogen fertilizers [41]. Similar benefits were seen when *Trichoderma* was co-inoculated with endophytes in drought-stressed plants [42]. Despite some concerns regarding its compatibility with arbuscular mycorrhizal fungi (AMF)—mainly due to its potential for parasitism, its volatile compound emissions, and competition for resources—studies have shown that *Trichoderma* and AMF can be used together effectively [43,44]. Co-application has resulted in improved crop yields, successful colonization in tomato seedlings, and even enhanced AMF levels in wheat rhizospheres [45]. In some cases, *Trichoderma* has supported AMF interactions in non-mycorrhizal hosts, like rapeseed [46].

Nevertheless, compatibility assessments should be performed on a case-by-case basis, considering that *Trichoderma* generally colonizes roots more rapidly than mycorrhizal fungi. Some *Trichoderma* strains have been identified as harmful—for example, *T. aggressivum*, *T. pleuroti*, and *T.pleuroticola* have been reported to infect edible mushrooms [47] and *T. brevicompactum* produces trichodermin, a toxin with phytotoxic effects on tomato plants [48]. Moreover, *T. longibrachiatum*, which can thrive at elevated temperatures (37 °C), has been implicated in opportunistic infections in immunocompromised individuals. Importantly, such adverse traits are not found in the *Trichoderma* strains typically used in agriculture, as these undergo rigorous testing to ensure safety prior to commercialization. Interestingly, *Trichoderma* may also serve as a biological control agent against leaf-cutting ants by disrupting their fungal symbionts. Its endophytic presence in plant material transported by ants may function as a “Trojan horse”, ultimately benefiting the plant by reducing pest pressure [49].

## 3. The Multifaceted Role of *Trichoderma* in Enhancing Plant Growth and Stress Tolerance

*Trichoderma* stands as a crucial group of plant-growth-promoting fungi (PGPF) within rhizosphere soils, fundamentally contributing to enhanced plant development and stress tolerance [50]. As symbiotic microorganisms, *Trichoderma* coexist with plant root ecosystems, where they facilitate the solubilization of soil minerals, leading to improved nutrient availability and efficiency, thereby fostering plant growth [50]. These fungi induce systemic resistance in plants and produce a variety of growth-promoting compounds, significantly augmenting overall plant growth and development [51]. Their impact extends to multiple facets of plant physiology, including improvement in seed germination, viability, root development, and root structure, increased flowering, and enhanced yield quality—all crucial elements in optimizing the photosynthesis process [52].

In terms of species-specific effects, distinct strains such as *Trichoderma reesei*, *Trichoderma longibrachiatum*, and *Trichoderma harzianum* have been identified for their ability to expedite orchard grass growth and enhance its nutritional content, which points to their potential role in pasture and field applications [53]. The multifaceted role of *Trichoderma* is further exemplified by their production of volatile organic compounds (VOCs) like 2-heptanone, 2-pentyl furan (2-PF), and 6-pentyl-2H-pyran-2-one (6-PP), which function as crucial agents in improving various physiological and developmental aspects of plants. These compounds have been shown to enhance aerial and root dry weight, directly influencing the primary growth direction of species such as tomatoes and *Arabidopsis thaliana* [54,55,56].

Furthermore, *Trichoderma* produce secondary metabolites like harzianolide, demonstrated to promote growth in tomato seedlings across different growing conditions, including natural soil and hydroponic systems [57,58]. Beyond these growth enhancements, *Trichoderma atroviride* and *Trichoderma virens* have been revealed to generate indole acetic acid (IAA) and other auxin-related compounds [28,59]. IAA plays a pivotal role in early root development, notably during the initial stages of plant maturity. The interaction of *Trichoderma* with plant signaling compounds underlines its potential to act as an alternative biofertilizer that minimizes the ecological impact of conventional chemical fertilizers.

This expansive role in promoting plant growth is matched by the ability of *Trichoderma* to bolster plant tolerance to both biotic and abiotic stressors. Of particular note is their ability to mitigate the impacts of common agricultural stresses such as drought, salinity, extreme temperatures, and heavy metal deposition, all of which significantly affect crop output [52,60,61,62,63]. In adverse environmental conditions, *Trichoderma* can enhance antioxidant defense systems, reducing reactive oxygen species (ROS) accumulation and thereby mitigating tissue damage and maintaining cellular integrity [64,65]. The distinct antioxidant system of *Trichoderma* not only protects its own genomic stability by eliminating ROS but also supports plant resilience against environmental challenges [26,32].

Furthermore, studies have shown that application of *Trichoderma* enhances plant resistance through up-regulation of stress-responsive genes associated with hormonal pathways like ethylene and abscisic acid, which are crucial for plants coping with stress conditions [66,67]. Specific strains such as *Trichoderma parareesei* have been demonstrated to enhance rapeseed plant resistance to salinity and drought stress through these mechanisms, significantly boosting yield compared with non-inoculated plants [61]. This adaptive capability signifies *Trichoderma*’s potential in developing resilient crop varieties capable of thriving in challenging environmental conditions.

The exploration of *Trichoderma* in agricultural practices is increasingly prominent due to its effectiveness in enhancing plant growth and development, coupled with its ability to serve as a biological control agent against various pathogens. The discovery and application of *Trichoderma*-derived products as biofertilizers align with sustainable agriculture goals, offering a viable means to improve crop productivity while mitigating the environmental damage associated with traditional fertilizers [65,68].

## 4. *Trichoderma* Biocontrol in Crop Protection

*Trichoderma* is a versatile biocontrol agent (BCA) due to its ability to directly antagonize target organisms and stimulate plant defense responses against biotic stressors. While its direct biocontrol action has been well-documented, it is important to recognize that not all *Trichoderma* species or strains exhibit uniform efficacy [69]. Their effectiveness varies across crops, geographic locations, and environmental conditions. Strains of interest in agriculture belong primarily to two groups, ST (e.g., *T. atroviride*, *T. gamsii*, and *T. viride*) and HV (e.g., *T. harzianum* and *T. afroharzianum*) [70]. The success of these strains depends on their biological characteristics—such as rapid growth, prolific sporulation, and the ability to colonize diverse environments—as well as their biochemical abilities, including enzymes that degrade host cell walls, secondary metabolites, and VOCs such as methyl jasmonate that modulate the host plant’s physiological responses, shape the composition of microbial communities in the rhizosphere, and ultimately alter the efficacy of natural defenses against phytopathogens [54,71].

*Trichoderma* employs five primary mechanisms for direct antagonism (Figure 2) [72]: Parasitism (*Trichoderma* acts as a mycoparasite, feeding on fungal pathogens), Antibiosis (secondary metabolites inhibit pathogen growth and microbial competition), Enzymatic Activity (enzymes like chitinases target nematodes and pests), Competition (*Trichoderma* competes for resources like nutrients and space, promoting its colonization), and VOCs (VOCs attract natural predators of insect pests).

In addition to these direct mechanisms, *Trichoderma* induces plant immune responses that help protect against both biotic and abiotic stresses. *Trichoderma* has proven effective in controlling economically important phytopathogens such as *Botrytis*, *Fusarium*, *Rhizoctonia*, *Pythium*, and *Phytophthora*. Secondary metabolites can also inhibit bacterial pathogens like *Pseudomonas* and *Xanthomonas* [73]. The genes linked to mycoparasitism and enzyme production have been shown to inhibit spore germination, hyphal growth, and fungal structures in numerous pathogens [74].

*Trichoderma* species are prolific producers of over 120 distinct secondary metabolites, encompassing diverse chemical classes such as terpenes (e.g., trichodermin and harzianopyridone), pyrones (e.g., 6-pentyl-2H-pyran-2-one), and nonribosomal peptides (e.g., gliotoxin and peptaibols), many of which exhibit potent antibiotic properties [30]. These compounds disrupt the growth and viability of a wide array of phytopathogens, including fungi (such as *Fusarium* and *Rhizoctonia* spp.), oomycetes (like *Phytophthora* and *Pythium* spp.), and bacteria (e.g., *Pseudomonas* and *Erwinia* spp.), by mechanisms such as inhibiting cell wall synthesis, permeabilizing membranes, or interfering with essential metabolic pathways like protein synthesis and respiration [71]. Notably, when purified and applied in isolation, these secondary metabolites can replicate the biocontrol efficacy observed in living *Trichoderma* cultures, often achieving pathogen inhibition rates comparable to whole-organism inoculants in controlled assays [75]. Despite being considered necrotrophic, *Trichoderma* may enter its prey via holes in the cell wall, utilizing a hemibiotrophic parasitic strategy that causes less extensive damage.

*Trichoderma* is also known for its suppressive effects on nematodes, such as Meloidogyne root-knot nematodes, and other types like *Heterodera* and *Pratylenchus*. It suppresses nematode eggs through proteases and chitinases or inhibits egg hatching with secondary metabolites [76]. Additionally, *Trichoderma* directly controls insects through enzymatic activity and inhibits cuticle formation, with secondary metabolites showing inhibitory effects on insect larvae [77].

As a competitor, *Trichoderma* counters pathogen strategies by secreting proteases that inhibit enzymes used by pathogens to invade plant tissues. Secondary metabolites can also downregulate pathogen virulence genes, such as those modulated by *Trichoderma arundinaceum* to reduce *Botrytis cinerea*’s virulence. Finally, *Trichoderma* produces VOCs, such as 6PP, which attract parasitoids and predators of insect pests, enhancing plant defense [78].

*Trichoderma* acts as an indirect biocontrol agent (BCA) by stimulating plant immune responses, leading to quicker and more robust activation of defense mechanisms upon subsequent stimuli (Figure 2). This process, known as “priming”, provides long-term protection by modulating various phytohormone-dependent pathways. Priming plays a crucial role in indirect biocontrol, as both types of responses share similar origins and development, despite being triggered by different stimuli. The interactions between *Trichoderma* and plants, and the mechanisms of signal activation and systemic transmission, have been extensively reviewed. Structural components of *Trichoderma*’s cell wall and membrane, such as chitin, β-glucans, and sterols, act as microorganism-associated molecular patterns (MAMPs). When recognized by plant pattern recognition receptors (PRRs), these MAMPs initiate MAMP-triggered immunity, which is more robust than pathogen-triggered immunity, thereby enhancing plant resistance. Upon reaching plant roots, *Trichoderma* prompts an increase in salicylic acid levels, a key phytohormone that controls early root colonization, confining *Trichoderma* to the apoplastic spaces of the epidermis and cortex. *Trichoderma* further enhances plant immunity through effector proteins and metabolites like xylanase EIX, LysM protein Tal6, ceratoplatanin Sm1, peptaibol alamethicin, and terpenes such as trichodiene and harzianum A, contributing to effector-triggered immunity even without PRR interactions. These secreted effectors, along with *Trichoderma*’s ability to tolerate reactive oxygen species (ROS), facilitate endophytic colonization and a non-harmful relationship with the plant, priming its defenses just below the threshold for effective resistance. The precise interactions between cytoplasmic nucleotide-binding site leucine-rich repeat (NLR) receptors and *Trichoderma*’s effectors remain only partially understood. In tomato plants, NLR receptors are notably more abundant in roots inoculated with *T. atroviride* and *Rhizoctonia solani*, with similar responses observed in other *Trichoderma* strains [79]. These interactions are under further investigation, including the potential role of small RNA trafficking across kingdoms. Salicylic-acid-dependent defenses that limit early *Trichoderma* growth can spread throughout the plant, creating systemic acquired resistance (SAR) effective against biotrophic pathogens [80]. *Trichoderma* suppresses salicylic acid defenses and induces jasmonic acid biosynthesis, activating jasmonic-acid-responsive genes in root cells. This triggers jasmonic acid–ethylene-dependent induced systemic resistance (ISR), particularly effective against necrotrophic pathogens and herbivore attacks. *Trichoderma* exploits the antagonistic relationship between salicylic acid and jasmonic acid to colonize roots, as seen in *T. asperellum*, which accumulates high levels of jasmonic acid and ethylene within 24 h, supporting the role of priming in induced systemic resistance. However, research indicates that *Trichoderma*-induced defenses against fungi and viruses are modulated by both jasmonic acid–ethylene and salicylic acid. Beyond enhancing plant fitness, *Trichoderma* helps prevent nematode access to roots [81]. In tomato roots affected by root-knot nematodes (RKNs), *Trichoderma* adjusts plant immunity by modulating salicylic-acid- and jasmonic-acid-dependent defenses based on the nematode infection stage [82]. Additionally, *Trichoderma* primes defenses in both leaves and roots through small-RNA-mediated gene silencing and by inducing transcription of genes involved in RNA-dependent DNA methylation, fine-tuning the expression of salicylic acid and jasmonic acid–ethylene defense genes [59]. *Trichoderma* also activates plant systemic defenses through the release of volatile organic compounds (VOCs), triggering an oxidative burst effective against aphids. It can enhance the expression of genes producing protective enzymes against moths and modify plant metabolic pathways to produce phytochemicals that deter herbivores or disrupt the insect gut proteome, reducing feeding [83,84,85]. VOCs released by *Trichoderma* can attract parasitoids and predators of aphids, further strengthening plant defense. By modulating plant defenses and growth, *Trichoderma* helps mitigate the effects of unfavorable environmental conditions. Abiotic-stress-related phytohormones share regulatory pathways with MAMPs and damage-associated molecular patterns (DAMPs). However, plants often face a combination of biotic and abiotic stresses, leading to varied sensitivity and signal transduction. In such cases, *Trichoderma* regulates calcium ion (Ca^2+^) signaling, crucial for plant immunity under stress, helping plants adapt to environmental changes by prioritizing growth and stress response functions [86]. *Trichoderma* has been shown to improve plant drought tolerance, enhance antioxidant defense, and delay water deficit responses. Studies report that *Trichoderma* can modulate ROS scavenging and influence ethylene levels, aiding plants in coping with waterlogging, salinity, and other stress factors [87]. Furthermore, *Trichoderma* increases plant growth and salt tolerance, either through direct contact or VOCs. However, combining *Trichoderma* with inorganic fertilizers in salt-stressed plants may disrupt the phytohormone network, hindering effective adaptation to conflicting signals. The interaction between *Trichoderma* and plants is dynamic, with the expression of salicylic acid and jasmonic acid–ethylene-dependent defense genes fluctuating in response to biotic and abiotic stresses [88]. Over time, the effects of *Trichoderma* priming may diminish, but plants retain a “transcriptional memory” of previous priming events, allowing for stronger defense responses upon subsequent stimuli. This priming ability can be passed on to the next generation, resulting in offspring with enhanced resistance to pests and diseases.

## 5. Agricultural Applications of *Trichoderma*

Agricultural policies have evolved significantly since the 2015 UN Sustainable Development Goals, which emphasized the need for sustainable practices in food and agriculture, including pesticide and fertilizer management [89]. Climate change and intensive farming have led to biodiversity loss, pest spread, and chemical contamination of the environment, all of which impact both ecosystems and human health. In response, there has been a global shift toward reducing synthetic chemicals, increasing the role of beneficial microorganisms like *Trichoderma*. This shift has been reflected in *Trichoderma*’s growing use as a biological alternative in products such as biostimulants, bioprotectants, and biofertilizers.

*Trichoderma* is widely recognized as a biocontrol agent (BCA), used to control plant diseases through various mechanisms [65]. It has gained significant traction, with BCA registrations rising from 21 in 2014 to 144 globally in 2022, involving multiple strains across 40 countries [90]. Despite this, the registration process varies across nations, such as in India and China, where government research institutes test *Trichoderma* strains for controlling plant diseases. While some regions market *Trichoderma* as a biostimulant, it is not yet universally recognized as such, particularly in Europe. As the scientific community supports *Trichoderma*’s dual role as both a biocontrol agent and biostimulant, regulators must update policies to recognize its broad applications for more sustainable agricultural practices.

To boost the success and wider adoption of biological products in eco-sustainable agriculture, improving their shelf life, efficacy, and standards to match those of conventional chemicals is essential. Key strategies include enhancing the production of *Trichoderma* spores and improving their stress tolerance. An ideal *Trichoderma* product should select strains with strong biocontrol potential, the ability to colonize roots and promote plant growth or disease resistance.

Microbial consortia that include *Trichoderma* and other beneficial organisms, such as biocontrol bacteria and fungi, offer improved efficacy in agricultural applications [91]. These consortia may also include bioactive compounds from various microbial or botanical sources that enhance biocontrol and growth-promoting effects. Additionally, the development of new biofertilizers and probiotics using *Trichoderma*-based microbiomes can restore beneficial soil microbiota, especially in degraded agricultural areas.

As global agriculture increasingly shifts toward sustainable practices to minimize environmental impacts, reduce chemical residues in food chains, and combat issues like soil degradation and pesticide resistance, the careful selection of *Trichoderma* strains with high tolerance to residual agrochemicals—such as fungicides (e.g., triazoles or strobilurins), herbicides (e.g., glyphosate), and insecticides (e.g., neonicotinoids)—will be paramount. This tolerance ensures that *Trichoderma* can maintain its biocontrol efficacy and plant-growth-promoting activities even in transitional farming systems where chemical inputs are being phased out gradually, preventing disruptions to microbial community dynamics and avoiding unintended suppression of beneficial fungi [92,93]. Industrial processes and innovations like encapsulation and nanoparticle technology can improve the delivery and effectiveness of *Trichoderma* in large-scale applications.

*Trichoderma* is integral to agriculture’s transition toward a green and circular economy, focusing on reducing environmental impacts and recycling waste into valuable products. Biotechnological advancements may also enable *Trichoderma* gene expression in plants to improve resistance to pathogens and environmental stress, reducing the need for agrochemicals and enhancing crop resilience. Ongoing research into *Trichoderma* explores its applications in improving crop production in marginal lands, climate change resilience, bioremediation, and reducing greenhouse gas emissions.

## 6. Conclusions

In this review, we have explored *Trichoderma*’s standout roles as a natural biocontrol agent and plant growth booster, helping farmers protect crops and boost yields in real-world settings. Recent advances in classifying *Trichoderma* species are making it easier to pick the right ones for biotech applications, like developing targeted biofungicides. We have also gained better insights into how *Trichoderma* teams up with plant microbiomes and soil life, leading to healthier plants, improved soil quality, and even benefits for human and environmental health. Cutting-edge tools have revealed how it kickstarts plant defenses, providing built-in shields against diseases, pests, and stresses like drought or poor soil—practical perks that can cut down on chemical use.

*Trichoderma*’s adaptability lets it thrive in all sorts of environments, from fields to greenhouses. The big opportunity? Leveraging its diversity for smarter, long-lasting farming strategies that ramp up production while keeping things eco-friendly. For instance, in organic systems, *Trichoderma* formulations have shown promise in controlling pests like *Fusarium* in tomatoes or nematodes in bananas, potentially slashing pesticide needs by 30–50% based on field trials. It could also help tackle climate challenges, secure food supplies, and shift agriculture toward greener alternatives.

That said, rolling out *Trichoderma* products is not straightforward—regulatory hurdles, inconsistent performance in varying conditions, and gaps in collaboration between scientists, farmers, and regulators slow things down. Yet, with global policies pushing for sustainable farming and “One Health” approaches, there is momentum to overcome these.

Looking ahead, let us think big. Multidisciplinary efforts could engineer *Trichoderma* strains via CRISPR for supercharged traits, like enhanced drought tolerance in staple crops. Pairing it with precision tech, such as drone-applied bioinoculants or microbiome sensors, might revolutionize organic farming. By addressing scalability and policy barriers, we can turn *Trichoderma* into a go-to tool for resilient, chemical-free agriculture—ultimately leading to bountiful harvests, healthier ecosystems, and a better quality of life for all.

## Figures and Tables

**Figure 1 microorganisms-13-01840-f001:**
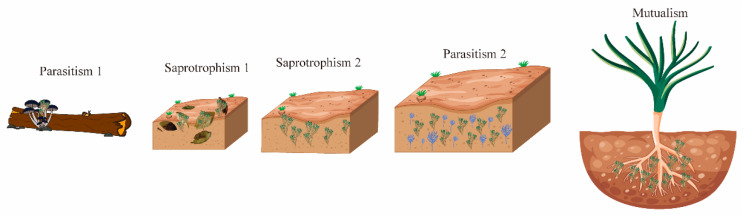
*Trichoderma* and the plant microbiome: allies in stress, complexity in coexistence. The evolutionary trajectory of *Trichoderma* spp. reveals key ecophysiological shifts. Parasitism 1: ancient mycoparasitism targeting wood-decomposing *Basidiomycota*. Saprotrophism 1: transition to saprotrophism during the Cretaceous–Paleogene extinction, thriving on dead plant material via acquired carbohydrate hydrolytic enzymes. Saprotrophism 2: adaptation as soil-colonizing saprophytes. Parasitism 2: expanded parasitism to *Ascomycota*, oomycetes, basidiomycetes, and nematodes. Mutualism: rhizosphere-driven shift to mutualism with plants, culminating in endophytism as the latest development.

**Figure 2 microorganisms-13-01840-f002:**
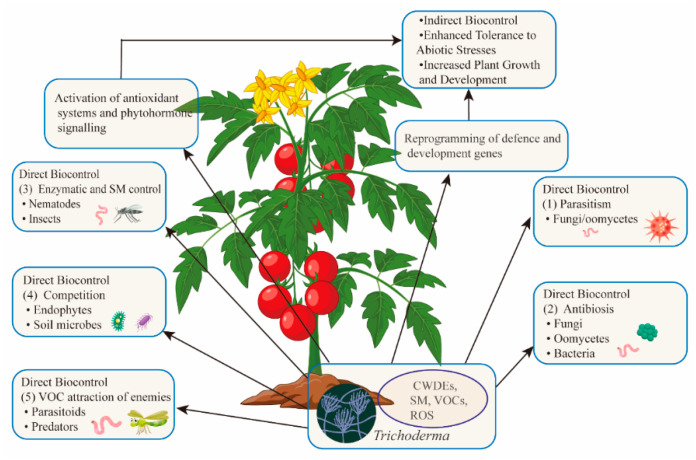
*Trichoderma*: a multifunctional ally for crops. *Trichoderma* benefits plants by directly antagonizing pathogens and pests, altering soil conditions, and attracting natural enemies of pests. It also boosts plant defenses through immune activation and promotes growth. Overall, *Trichoderma* enhances plant health and yields.

## Data Availability

No new data were created or analyzed in this study. Data sharing is not applicable to this article.

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
