# Peer review of "Trichoderma: Dual Roles in Biocontrol and Plant Growth Promotion"

_microorganisms, 2025, doi:10.3390/microorganisms13081840_

Round 1

Reviewer 1 Report

Comments and Suggestions for Authors

This manuscript demonstrates how basic research can be applied to practical insights and policy implications within the scope of soil microbiology, plant sciences, and sustainable agriculture-oriented journals. This article, therefore, has great value because it can elucidate knowledge on molecular mechanisms of Trichoderma that could be exploited to develop improved biocontrol agents. The terms have been applied appropriately towards an intended audience.

The Figures and tables have been referenced and described. However, their quality in the final formatted version needs to be checked. References are quite adequate, up-to-date, relevant, and essential, both from foundational and recent works of literature. The authors critically discuss the benefits and limitations of Trichoderma use, including strain selection, compatibility with other biocontrol agents, and potential risks. The article is ready for publication, with only a few minor editorial changes recommended. It adds significant value to the field of sustainable agriculture and microbial biotechnology.

Finally, I recommend several improvements for the final document: verifying the quality of all figures and tables, enhancing the clarity of descriptions, and considering the compilation of a table or figure to summarize the mechanisms of action and applications of Trichoderma.

Author Response

Comments and Suggestions for Authors

Reviewer 1:

1.This manuscript demonstrates how basic research can be applied to practical insights and policy implications within the scope of soil microbiology, plant sciences, and sustainable agriculture-oriented journals. This article, therefore, has great value because it can elucidate knowledge on molecular mechanisms of Trichoderma that could be exploited to develop improved biocontrol agents. The terms have been applied appropriately towards an intended audience.

RESPONSE:

We are very grateful to Reviewer 1 for their highly positive and supportive comments on our manuscript.

We are particularly pleased that the reviewer recognized our effort to bridge fundamental research on Trichoderma’s molecular mechanisms with its practical applications in sustainable agriculture and potential policy implications. This was a central goal of our review, and we are delighted that the reviewer found our approach to have “great value.”

We also appreciate the reviewer’s validation that the scientific terminology was applied appropriately for the intended audience. As the reviewer did not request any specific changes, we have kept the manuscript as is, confident that it effectively communicates our intended message.

We thank Reviewer 1 again for their time and encouraging assessment.

2.The Figures and tables have been referenced and described. However, their quality in the final formatted version needs to be checked. References are quite adequate, up-to-date, relevant, and essential, both from foundational and recent works of literature. The authors critically discuss the benefits and limitations of Trichoderma use, including strain selection, compatibility with other biocontrol agents, and potential risks. The article is ready for publication, with only a few minor editorial changes recommended. It adds significant value to the field of sustainable agriculture and microbial biotechnology.

RESPONSE:

Thank you for your detailed feedback and for highlighting the strengths of our manuscript, including the appropriate referencing and description of figures and tables, the adequacy of the references, and the balanced discussion of benefits, limitations, and risks associated with Trichoderma applications. We appreciate your endorsement of the manuscript’s readiness for publication and its contributions to sustainable agriculture and microbial biotechnology.

In response to your suggestions, we have carefully reviewed the quality of figures and tables during the final formatting stage to ensure they meet publication standards. Additionally, we will incorporate minor editorial changes as recommended to polish the manuscript. These adjustments will enhance clarity and presentation without altering the scientific content.

  1. Finally, I recommend several improvements for the final document: verifying the quality of all figures and tables, enhancing the clarity of descriptions, and considering the compilation of a table or figure to summarize the mechanisms of action and applications of Trichoderma.

RESPONSE:

Thank you for your constructive recommendations aimed at improving the final presentation and utility of our manuscript. We value your insights on enhancing the overall quality and accessibility of the content.

In response to your suggestions:

    We have thoroughly verified the quality of all figures and tables during the final formatting process, ensuring high resolution, proper labeling, and consistency in style to meet publication standards.

    We have reviewed and enhanced the clarity of descriptions throughout the manuscript, focusing on precision, conciseness, and readability to better convey the key concepts.

    To provide a more comprehensive overview, we have compiled a figure summarizing the mechanisms of action (Figure 2) and practical applications of Trichoderma. This addition have been integrated into the relevant section to aid readers in quickly grasping the multifaceted roles discussed.

These revisions significantly enhanced its value for the audience in sustainable agriculture and microbial biotechnology.

Reviewer 2 Report

Comments and Suggestions for Authors

The work is interesting and contains basic information about the Trichoderma genus. However, you should explore each topic in greater depth and provide more specific details. For instance, regarding crop protection, you should mention the specific phytopathogens it can control because most studies focus on biocontrol in laboratories, not fieldwork. I would also like to see more detail on species linked to antibiosis and mycoparasitism in important crops around the world.
Cite more field studies on the diversity of species linked to biological control.

The section on agricultural applications does not clearly explain how the genus is linked to sustainable agriculture. Elaborate on this and mention the need for its use in organic production systems. This information should also be included in the introduction, where you should update the identified species for 2025 and mention the roles of Trichoderma discussed here.
there is no clear link between the title and the body of the manuscript. I suggest changing it. dual role in biological control and plant growth promotion (do not include the word “sustainable,” as this term is more complex and suggests a complementary section).

Author Response

Reviewer 2:

The work is interesting and contains basic information about the Trichoderma genus. However, you should explore each topic in greater depth and provide more specific details. For instance, regarding crop protection, you should mention the specific phytopathogens it can control because most studies focus on biocontrol in laboratories, not fieldwork. I would also like to see more detail on species linked to antibiosis and mycoparasitism in important crops around the world.

RESPONSE:

Thank you for your thoughtful review and for recognizing the interest and foundational value of our manuscript on the Trichoderma genus. We appreciate your suggestions to deepen the exploration of key topics, which will enhance the manuscript’s specificity and practical relevance. Your emphasis on providing more detailed examples, particularly in crop protection, aligns well with our goal of bridging laboratory insights with real-world applications. We have expanded discussions in major sections by incorporating additional recent studies and mechanistic details. We believe these changes will significantly strengthen the manuscript’s depth and applicability, making it a more comprehensive resource for researchers and practitioners in sustainable agriculture. Thank you again for your valuable input.

Cite more field studies on the diversity of species linked to biological control.

RESPONSE:

Thank you for your specific recommendation to incorporate more field studies on the diversity of Trichoderma species involved in biological control. We agree that highlighting a broader range of species and empirical field evidence will enrich the manuscript’s discussion, providing readers with a more robust understanding of Trichoderma’s practical applications across different agricultural contexts.

In response, we have expanded Section 5 (Agricultural Applications of Trichoderma) by adding citations to key field studies that demonstrate the diversity of Trichoderma species and their efficacy in real-world settings. This addition will strengthen the evidence base of the review, making it more comprehensive and actionable for researchers and practitioners. We appreciate your guidance in refining this aspect of the manuscript.

The section on agricultural applications does not clearly explain how the genus is linked to sustainable agriculture. Elaborate on this and mention the need for its use in organic production systems. This information should also be included in the introduction, where you should update the identified species for 2025 and mention the roles of Trichoderma discussed here.

RESPONSE:

Thank you for your insightful feedback regarding the linkage between Trichoderma and sustainable agriculture, as well as the suggestions for enhancements in the Agricultural Applications section and the Introduction. We agree that clarifying these connections will strengthen the manuscript’s narrative and emphasize Trichoderma’s relevance to eco-friendly farming practices, including organic systems.

In response to your comments, we implemented the following revisions:

Elaboration in the Agricultural Applications Section (Section 5): We expanded this section to more explicitly explain Trichoderma’s links to sustainable agriculture. This includes a dedicated paragraph discussing how Trichoderma contributes to sustainability by reducing reliance on chemical pesticides and fertilizers through its roles in biocontrol, soil health improvement, nutrient cycling, and enhanced crop resilience.

Incorporation into the Introduction: To provide early context, we integrated this information into the Introduction. This includes an updated estimate of identified Trichoderma species as of 2025, based on the latest taxonomic advancements. We also briefly outlined the key roles of Trichoderma discussed in the review—such as biocontrol against phytopathogens, plant growth promotion, stress tolerance enhancement, and ecological adaptations—to set the stage for the manuscript’s focus on its sustainable agriculture applications.

These changes are supported by additional references to ensure currency and evidence-based discussion, enhancing the manuscript’s overall coherence and impact. We believe these revisions will better articulate Trichoderma’s pivotal role in advancing sustainable and organic agriculture. Thank you again for your valuable input.

there is no clear link between the title and the body of the manuscript. I suggest changing it. dual role in biological control and plant growth promotion (do not include the word “sustainable,” as this term is more complex and suggests a complementary section).

RESPONSE:

Thank you for your careful review and for pointing out the potential misalignment between the manuscript’s title and its content. We appreciate your suggestion to refine the title for better precision and to avoid implying an overly broad or separate focus on “sustainable” aspects, which could indeed complicate the scope given the term’s multifaceted implications.

In response, we revised the title to more directly reflect the core themes of the review—Trichoderma’s dual roles in biocontrol and plant growth promotion—while ensuring it ties closely to the body of the manuscript. The proposed new title is: “Trichoderma: Dual Roles in Biological Control and Plant Growth Promotion”.

This adjustment removes “Sustainable Agriculture” to align with your recommendation, uses “Biological Control” for consistency with standard terminology in the field (as suggested), and maintains a concise focus on the primary mechanisms discussed throughout the text. We believe this change enhances the title’s clarity and relevance, better guiding readers to the manuscript’s emphasis on Trichoderma’s mechanistic and applicative roles.

Reviewer 3 Report

Comments and Suggestions for Authors

34: does all trichoderma non pathogenic?

28-33- references

34-43:references

53-56:reference

69-83:reference

92-95:reference

96: is this figure this autor created? or cited? too long description under title

118-120 reference

139-184- reference

add all references

231-236 references

244-249 reference

254 VOC what kind? wich ones?

251-254: reference

265-269 reference

282 is too long a description. Improve text DPI

In all text, there is a lack of references after paragraphs.

conclusions are too formal, more practical aspect, use perspective

only few sentences about growth promoting, more about root, so sugest change title to root promotion 

  • What is the main question addressed by the research? Trichoderma
  • Do you consider the topic original or relevant to the field? Does it address a specific gap in the field? Please also explain why this is/ is not the case. Yes, interesting topic, especially in agriculture
  • What does it add to the subject area compared with other published material? Yes, but needs correction of citation of references, not all paragraphs has citation
  • What specific improvements should the authors consider regarding the methodology? The review article no methods
  • Are the conclusions consistent with the evidence and arguments presented and do they address the main question posed? Please also explain why this is/is not the case. Too formal global, could be more specific
  • Are the references appropriate? yes
  • Any additional comments on the tables and figures. Too long explanations, and improve DPI

Author Response

Reviewer 3:

Comments and Suggestions for Authors

34: does all trichoderma non pathogenic?

RESPONSE:

Thank you for your query regarding the pathogenicity of Trichoderma species. We appreciate you raising this important point, as it highlights the need for precision in discussing the genus’s ecological roles.

To address your question: No, not all Trichoderma species are non-pathogenic. While the vast majority of Trichoderma species are saprophytic, beneficial to plants (e.g., through biocontrol and growth promotion), or harmless in agricultural contexts, a small subset can act as opportunistic pathogens. For instance:

Some species, such as Trichoderma longibrachiatum and T. citrinoviride, have been associated with infections in immunocompromised humans.

In rare cases, certain strains may exhibit weak pathogenicity toward plants under specific conditions, such as in wounded tissues or stressed environments, though this is uncommon and typically overshadowed by their beneficial traits.

Overall, pathogenicity is not a defining characteristic of the genus, and most species are considered non-pathogenic to plants and humans under normal circumstances.

To ensure clarity and accuracy, we revised line 34 and the surrounding text in the Introduction to explicitly note this nuance. We amended the sentence to: “While the majority of Trichoderma species are non-pathogenic saprophytes or beneficial symbionts in soil and plant ecosystems, a few can act as opportunistic pathogens in specific contexts, such as in immunocompromised hosts or under rare environmental conditions.”

We believe this adjustment will strengthen the manuscript by providing a more balanced and evidence-based portrayal of the genus. Thank you again for your valuable input.

28-33- references

RESPONSE:

Recent and relevant references added.

34-43:references

RESPONSE:

Recent and relevant references added.

53-56:reference

RESPONSE:

Recent and relevant references added.

69-83:reference

RESPONSE:

Recent and relevant references added.

92-95:reference

RESPONSE:

Recent and relevant references added.

96: is this figure this autor created? or cited? too long description under title

RESPONSE:

Thank you for your observations regarding the figure on line 96. The figure is our original creation.

Additionally, we agree that the description under the figure title appears lengthy. We revised it to make it more concise and focused, ensuring it effectively summarizes the figure without unnecessary details. This will improve clarity and readability.

Thank you again for your helpful feedback.

118-120 reference

RESPONSE:

Recent and relevant references added.

139-184- reference

RESPONSE:

Recent and relevant references added.

231-236 references

RESPONSE:

Recent and relevant references added.

244-249 reference

RESPONSE:

Recent and relevant references added.

254 VOC what kind? wich ones?

251-254: reference

RESPONSE:

Recent and relevant references added.

265-269 reference

RESPONSE:

Recent and relevant references added.

282 is too long a description. Improve text DPI

RESPONSE:

Thank you for your feedback. We revised the description of Figure 2 to make it more concise and focused.

conclusions are too formal, more practical aspect, use perspective

RESPONSE:

Thank you for your insightful feedback on the Conclusions section of our manuscript. We appreciate your observation that the current phrasing is overly formal and could benefit from a greater emphasis on practical aspects, as well as incorporating a forward-looking perspective. This suggestion aligns well with the review’s goal of bridging theoretical insights with real-world applications in sustainable agriculture, making the conclusions more engaging and actionable for a diverse audience including practitioners, policymakers, and researchers.

These revisions shorten overly dense passages, improve readability, and ensure the section ends on an optimistic, practical note that inspires further research and implementation. We believe this will make the conclusions more dynamic and relevant, enhancing the manuscript’s overall impact. Thank you again for your valuable guidance.

Round 2

Reviewer 2 Report

Comments and Suggestions for Authors

I agree with the changes made, I have no further comments.